# Extracellular Vesicle Proteome in Prostate Cancer: A Comparative Analysis of Mass Spectrometry Studies

**DOI:** 10.3390/ijms222413605

**Published:** 2021-12-19

**Authors:** Rui Miguel Marques Bernardino, Ricardo Leão, Rui Henrique, Luis Campos Pinheiro, Prashant Kumar, Prashanth Suravajhala, Hans Christian Beck, Ana Sofia Carvalho, Rune Matthiesen

**Affiliations:** 1Computational and Experimental Biology Group, Chronic Diseases Research Centre (CEDOC), NOVA Medical School, Faculdade de Ciências Médicas, Universidade NOVA de Lisboa, 1169-056 Lisboa, Portugal; ana.carvalho@nms.unl.pt; 2Urology Department, Centro Hospitalar e Universitário de Lisboa Central, 1169-050 Lisbon, Portugal; luis.campos@nms.unl.pt; 3Faculty of Medicine, University of Coimbra, 3000-370 Coimbra, Portugal; ricardo.leao@jmellosaude.pt; 4Pathology Department, Instituto Português de Oncologia, 4200-072 Porto, Portugal; henrique@ipoporto.min-saude.pt; 5Institute of Bioinformatics, International Technology Park, Bangalore 560066, India; prashant@ibioinformatics.org; 6Somaiya Institute of Research and Consultancy (SIRAC), Somaiya Vidyavihar University (SVU), Vidyavihar, Mumbai 400077, India; 7Amrita School of Biotechnology, Amrita Vishwa Vidyapeetham, Amritapuri Campus, Clappana P.O., Kollam 690525, India; prash@bioclues.org; 8Centre for Clinical Proteomics, Department of Clinical Biochemistry and Pharmacology, Odense University Hospital, 5000 Odense, Denmark; hans.christian.beck@rsyd.dk

**Keywords:** extracellular vesicles (EVs), urine extracellular vesicles (uEVs), prostate cancer, proteomics, biomarkers

## Abstract

Molecular diagnostics based on discovery research holds the promise of improving screening methods for prostate cancer (PCa). Furthermore, the congregated information prompts the question whether the urinary extracellular vesicles (uEV) proteome has been thoroughly explored, especially at the proteome level. In fact, most extracellular vesicles (EV) based biomarker studies have mainly targeted plasma or serum. Therefore, in this study, we aim to inquire about possible strategies for urinary biomarker discovery particularly focused on the proteome of urine EVs. Proteomics data deposited in the PRIDE archive were reanalyzed to target identifications of potential PCa markers. Network analysis of the markers proposed by different prostate cancer studies revealed moderate overlap. The recent throughput improvements in mass spectrometry together with the network analysis performed in this study, suggest that a larger standardized cohort may provide potential biomarkers that are able to fully characterize the heterogeneity of PCa. According to our analysis PCa studies based on urinary EV proteome presents higher protein coverage compared to plasma, plasma EV, and voided urine proteome. This together with a direct interaction of the prostate gland and urethra makes uEVs an attractive option for protein biomarker studies. In addition, urinary proteome based PCa studies must also evaluate samples from bladder and renal cancers to assess specificity for PCa.

## 1. Introduction

Prostate cancer (PCa) is mostly an asymptomatic and slowly growing tumor, which might start developing in young men, but is typically only possible to start detecting around the age of 40–50 with an average age of about 66 years [1]. PCa is the most frequently diagnosed cancer in 112 countries and the second leading cause of cancer related death among men in developed countries, with an estimated 1.4 million new cases and 375,000 deaths worldwide in 2020 [2,3,4]. Its incidence is increasing and is highest in Japan and the USA [5]. Current PCa screening methods consist of serum prostate-specific antigen (PSA), digital rectal examination (DRE) which upon suspected PCa is recommended for Magnetic Resonance Imaging (MRI) or transrectal ultrasound (TRUS) analysis. Despite many controversies, serum prostate-specific antigen (PSA) measurement remains the most widely used tool for PCa detection. Nevertheless, PSA measurement performance is far from ideal (Figure 1). PSA has only 25–40% positive predictive value for PCa detection, and eventually 65–70% of men with PSA serum levels between 4.0 and 10.0 ng/mL present a negative prostate biopsy [6]. Additionally, up to 15% of men with PCa have PSA levels below 4.0 ng/mL. Both serum PSA and digital rectal examination (DRE) present low specificity for diagnosis. As such, the lack of a specific marker(s) for PCa diagnosis with a capacity to distinguish indolent from aggressive disease leads to unnecessary prostate biopsies and unnecessary treatments. Furthermore, a subset of clinically significant PCa (csPCa) will be left undetected [7].

Additionally, standard transrectal ultrasound (TRUS) is unreliable for detecting PCa [8,9]. Consequently, ultrasonography–guided biopsy of the prostate is associated with under detection of higher-grade csPCa and the over detection of low-grade csPCa [10]. On the other hand, multiparametric magnetic resonance imaging (mpMRI) of the prostate allows higher accuracy and identification of significant lesions. Although achieving better accuracy, mpMRI findings still miss identification of the cribriform variant and tumors with predominantly cribriform pattern. Importantly, mpMRI is not highly sensitive at identifying International Society Urological Pathology (ISUP) Grade Group 1 (Gleason 6) lesions [11].

Options to overcome these limitations include targeted biopsy techniques such as mpMRI-guided biopsies. As mpMRI of the prostate progresses, better correlation with histology is expected, allowing for pre-biopsy identification of aggressive tumor patterns. The studies that have investigated the histologic correlation between mpMRI findings and a type of aggressive histologic growth (cribriform pattern) have shown conflicting results, as cribriform predominant tumors were often less visible on mpMRI than noncribriform predominant tumors [12]. Most MRI-guided biopsy misdiagnosis are due to errors in lesion targeting (51.2%), followed by MRI-invisible lesions (40.5%), and MRI lesions missed by the radiologist (7.1%) [13]. Despite higher accuracy of MRI-guided biopsy compared to standard TRUS biopsies, mpMRI targeted biopsies with collection of tissue of detected suspicious lesions, still presents limitations, leading to potentially under diagnosis of aggressive patterns, namely cribriform. Template mapping biopsy involves applying a template to insert several fine needles through the skin in the area between the scrotum and the anus (the perineum) into the prostate gland in order to obtain several tissue samples for testing. Template mapping biopsy is more accurate in assessing volume and grade of tumor compared to TRUS biopsy [14].

The above impediments of the current diagnosis methods call for improved early diagnostics, new methods for reducing over diagnosis and over treatment of insignificant PCa tumors. As such, biomarkers with increased reliability and accuracy are paramount for addressing these clinical problems.

The prostate lies in direct contact with the urethra, and secretes products that can be detected in urine and therefore constitute a liquid biopsy source of biomarkers for PCa [15,16,17]. Unfortunately, urine is a challenging fluid for the discovery of protein-based biomarkers due to the presence of certain salts, the low protein concentration and the extreme dynamic range of protein concentration [18].

Urinary extracellular vesicles (uEVs) overcome these challenges by concentrating proteins and at the same time lower highly abundant proteins such as serum albumin and uromodulin [19,20]. Furthermore, the isolation of EVs lowers concentration of salts in the sample. Moreover, EVs contain a rich source for prostate-derived products and can be isolated from urine. EV preparations typically contain various particles such as exosomes, microvesicles (MV), ectosomes, protein aggregates and large protein complexes [21,22]. In PCa studies, different nomenclature is applied to a mixture of different vesicles such as prostasomes, EVs, apoptotic bodies, tumor-derived MVs or exosomes. In this study, the term EV is applied throughout to a mixture of different subpopulations of various vesicles types. Exosomes are 50–150 nm sized membrane vesicles that are shed by many mammalian cell types, including malignant cells, which are formed within the endosomal network and released upon fusion of multivesicular bodies with the plasma membrane [23]. EVs have been considered a promising and easily accessible biomarker reservoir for various diseases, as their content (such as proteins, lipids, DNA, and RNA) is thought to reflect the molecular composition of their tissue of origin [24,25,26]. Recently, Zhang et al. described a sub population termed exomeres (~35 nm) isolated from cell culture media [27]. It is currently unclear if exomeres are detectable in liquid biopsies and if they have any clinical potential.

Relatively few reviews provide a focused discussion on urinary EVs’ application in PCa diagnosis (Table 1).

The literature review by Drake and Kislinger, 2014 [29], is the only one focusing on potential protein biomarkers for prostate cancer. Furthermore, at the publication time there were few high quality data sets available. Additionally, recent reviews tend to focus mainly on isolation methods for EVs.

Presently, no standardization of uEV studies exists, and this constitutes a need for possible improvement for future research. The goal of the Urine Task Force of the International Society for Extracellular Vesicles assembled nephrologists, urologists, cardiologists and biologists with active experience in uEV research to establish best practice and standardize the methods in the field. The Urine Task Force provides an excellent source of information on standardization and recommendations for uEVs [33].

Currently, approved molecular diagnostic tests either measure mRNA expression [28] or detection of TMPRSS2:ERG (T2:ERG) gene fusion [34]. ExoDx is the only test approved that measures PCA3 and ERG RNA expression in EVs from voided urine. None of the mentioned urinary-based diagnostic tests with approval is based on proteome biomarkers. On the contrary, serum based markers for PCa are based mainly on proteins such as tPSA, fPSA, p2PSA (a PCa-specific fPSA isoform) and kallikrein 2 (KLK2) [35,36]. As such, this raises the question whether urine proteome has been thoroughly explored, especially at the proteome level. In fact, most EV based biomarker studies have targeted plasma or serum [28]. Therefore, we aim to highlight strategies for urine biomarkers and particularly focus on the proteome of uEVs.

As such, we intend to discuss the benefits of liquid biopsy in comparison to tissue biopsy, exploring functions of EVs in PCa progression and summarize the role of EV proteins in diagnosis and prognosis of PCa from previous studies. A detailed evaluation of studies targeting different liquid biopsies and omics technologies for PCa diagnosis was performed. The results suggested that proteomics and metabolomics approaches are relatively unexplored for studying uEVs. Based on this analysis, we considered that the current number of original research studies in proteomics were well suited for a more detailed review and evaluation.

## 2. Functions of EVs in Cancer

Typically, MVs are released from the cell membranes and co-isolate with exosomes that originate from multivesicular bodies (MVBs) in EV preparations.

### 2.1. Exosome Biogenesis

Exosomes, distinctly from microvesicles, originate from the endosomal system [37]. Endocytosis of cell surface integral membrane proteins forming endocytic vesicles are rapidly targeted to a distinct membrane-bound endocytic organelle referred to as the early endosome (EE). At this compartment, sorting events serve to target internalized proteins and lipids between three fate decisions destining them either for recycling to the plasma membrane, degradation in lysosomes or delivery to the trans-Golgi network. Maturation of EE to late endosomes undergoes a process of multiple rounds of cargo sorting and intraluminal vesicle (ILV) biogenesis. Similarly to EE, late endosomes can fuse with the lysosome, generating an endolysosome compartment that provides a controlled acidic environment for the degradation of the cargo-loaded ILVs, or fuse with the plasma membrane to release the content of ILVs as extracellular vesicles known as exosomes (Figure 2) [38]. Exosomes contain DNA, miRNA, mRNA, glycans, soluble proteins, metabolites, and various other signaling molecules from parental cells but it lacks ribosomal RNA. Glycans and glycoproteins in EVs constitute potential biomarkers in the clinic for both stratification and prognosis of cancer biomarkers [39], especially since EVs contain known tumor-associated glycans [39]. The biological role and loading of genomic DNA (gDNA) in exosomes is poorly understood but potentially involves micronuclei [40]. However, EV DNA constitutes a relevant target for diagnostics.

### 2.2. Microvesicles

Microvesicle biogenesis is a process by which the direct outward blebbing and pinching of the plasma membrane release the nascent microvesicle into the extracellular space [41]. Alternative terms for microvesicles in the literature are shedding vesicles, ectosomes, oncosomes, shedding bodies, and microparticles. MVs, like exosomes, are considered promising targets for advanced diagnosis and therapy.

### 2.3. Function of EVs in PCa

EVs from the tumor microenvironment are important regulators for enhancing prostate cell survival (Figure 3), proliferation, angiogenesis and evasion of immune surveillance, all together contributing to PCa progression [42,43].

Carcinoma-associated fibroblasts (CAFs) derived EVs can transfer miRNAs into neighboring epithelial cells causing the exponential growth of PCa cells [44]. Cancer cells-derived EVs are also involved in the regulation of signaling pathways. Proto-oncogene tyrosine protein kinase (C-Src), insulin-like growth factor I receptor and focal adhesion kinase (FAK) are abundantly present in EVs [45]. PCa EVs are capable of directly targeting immune cells and inducing down-regulation of the NKG2D receptor on lymphocytes, thus promoting tumor immune evasion [46]. Furthermore, EV miRNAs can induce inflammatory responses on recipient cells through protein and miRNA transfer. Thereby, leading to enhanced tumor proliferation and migration, and mediating cellular reprogramming on recipient cells. EV miRNAs were also shown to be involved in fibroblast proliferation, angiogenesis, differentiation, and migration in PCa [47].

Additionally, EVs were demonstrated to confer drug resistance by transferring its genetic contents which allow the recipient cells to reprogram and develop resistance [48]. It is believed that chemotherapeutic drugs are exported via EVs [49]. Docetaxel was one of the first drugs studied, including the analysis of 22Rv1 and DU145 cell lines and its respective docetaxel-resistant variants 22Rv1RD and DU145RD, respectively. The research showed that EVs from the docetaxel-resistant cells transmitted docetaxel-resistance to previously drug-sensitive parent cell lines. Resistant cell’s secreted EVs carried substantial amounts of P-gp drug transporter suggesting, at least partially, their involvement in the mechanism of acquired resistance [50].

PCa cells display bone tropism, and cancer derived EVs have been shown to determine organotropism [51]. Serum EVs from metastatic PCa patients showed high contents of miR-21 and miR-141, which regulated osteoclastogenesis and osteoblastogenesis [52,53]. EVs isolated from the murine PCa cell line TRAMP-C1 significantly decreased fusion and differentiation of monocytic osteoclast precursors to mature into multinucleated osteoclasts. The authors suggest that EVs released from tumor cells in the tumor-bone interface function in the pathological regulation of bone cell formation in the metastatic site [54]. We may hypothesize that EVs potentially provide informative insight into metastatic disease as EVs in biofluids will represent all advanced metastatic cancer cells whether these cells are in the systemic circulation or from cells found at the site of metastasis. Moreover, melanoma derived EVs are capable of establishing a metastatic niche for ensuing circulating tumor cells [55].

For therapeutic purposes, EVs have gained great interest as therapeutic vectors to target cancer cells. They serve as carriers to deliver therapeutic agents to tumor cells, leading to effective tumor eradication, while minimizing the side effects caused by standard targeted therapies and chemotherapies [56]. In addition, they have the capacity to deliver different types of cargo and to target specific cells. In vitro and in vivo experiments, confirm efficiency of EV delivery of different therapeutic agents [57]. EVs are being pursued as intercellular vectors for RNA-based therapy (both miRNAs and siRNAs), with a documented efficacy in animal models of disease. For example, exosome delivery of the tumor suppressor miR-143, in mice caused in vivo suppression of PCa [58]. On the other hand, the EV surface proteins can shield the tumor targets thereby enabling tumor cells to escape from the immune system attack [59].

## 3. Current Diagnostic Methods and Advantages of Using Liquid Biopsies

PCa diagnosis is currently based on histopathological analysis of cells from prostate tissue biopsy. Morphologically, prostate carcinomas are particularly complex regarding their propensity to display multiple histological patterns within a single tumor (Figure 4) [60]. The presence of multiple, independent foci of prostatic adenocarcinoma within the same gland is a common finding. Moreover, the initial biopsy grade may not reflect that of the resultant prostate specimen, due to sub sampling that could result from both grade heterogeneity and multifocality [61].

Furthermore, TRUS biopsy is unable to reliably identify PCa and is particularly poor at sampling cancers in the anterior and apical locations, contributing to the under detection of clinically significant disease [62]. Similarly, up to 40% of cases classified by TRUS biopsy as low grade are in fact higher grade disease in surgical radical prostatectomy specimens [63].

Advances in prostate mpMRI have allowed for MRI-targeted biopsies of suspicious imaging findings [11,64,65]. MRI-targeted biopsies result in a higher rate of detection of high-grade cancers [9,10,64,66] with an improved detection of clinically significant cancers.

The ideal test for PCa detection should embrace several properties, namely, cost effectiveness, and minimally invasive with few side effects (Figure 1). Liquid biopsy has therefore emerged as a promising minimally invasive method to achieve a molecular profiling of PCa that could overcome these limitations. Liquid biopsy is considered homogenous in contrast to tissue biopsies for which sampling sites affect biomarker outcome. The main biological biomarkers used in liquid biopsy originate from circulating tumor cells (CTCs), circulating tumor DNA (ctDNA) and EVs, including exosomes.

Compared to blood, urine is a relatively easy biofluid to collect in large quantities in a noninvasive manner, with only uromodulin and human serum albumin as relatively abundant proteins [67,68]. The total protein coverage in our hands is typically 2–3 times higher in uEV samples compared to plasma and plasma EVs. We typically identify ~600–800 proteins in plasma, ~1000 proteins in plasma EVs and 2000–3000 proteins in uEVs from a single patient sample using state of the art mass spectrometry. Under healthy physiological conditions, urine contains few cells: either epithelial cells from the lining of the urogenital tract, or blood-derived, such as immune cells. EVs released by cells have appeared as a source of noninvasive biomarkers for different pathological conditions including prostatic diseases [69,70,71]. Biofluids are obtained by either noninvasive or invasive methods. Examples of noninvasive biofluids constitute urine, saliva, seminal fluid, breast milk and sweat. Examples of minimally invasive biofluids, which are typically extracted by a needle or aspiration, range from blood, cerebrospinal fluid, pleural effusion and bronchoalveolar lavage. In the context of PCa blood, urine, and seminal fluids account for the most studied biofluids. All of these fluids may contain EVs released by tumor cells with a set of specific tumor-related biomolecules that are protected from degradation by the EV membrane [72,73]. Notably, among identified potential PCa biomarkers are several proteins [74], lipids [75], RNAs [69] and microRNAs [76] present in EVs. The advantages of uEVs for proteomics studies are multiple, namely, removal of abundant urine proteins, concentration of low abundant proteins, increased resistance to proteases and nucleases, and long term proteome stability at −80 °C storage. Furthermore, the noninvasive sampling opens the possibility for longitudinal studies. The disadvantages of uEVs compared to direct urine are increased processing time, increased cost, and increased variability due to additional isolation steps.

## 4. Overview of Liquid Biopsies and Omics Technologies Used in PCa Studies

The four main omics technologies applied in biomarker discovery are genomics, transcriptomics, proteomics and metabolomics. The different types of relevant biopsy samples applied to PCa are prostate tissue, plasma, serum, seminal fluids, urine, and expressed prostatic secretions (EPS) in urine. EPS in urine are collected in voided urine after DRE [29,77]. EVs isolated from EPS urine contain exosome components found in urine and prostasomes secreted from the prostate. EVs from tissues other than the urinary and male reproductive tract are also present in urine. In theory, EVs should not pass the glomerular filtration barrier (GFB) and basement membrane of the kidney (6 nm in the healthy state). Nonetheless, the kidneys filtration properties are likely jeopardized in pathological states thereby enabling EV passages [33,78,79]. However, EPS urine is enriched in EVs from the urinary and male reproductive tract. EPS urine furthermore contains cells from the urinary and male reproductive tract making it an interesting target for biopsy. Detection of PCA3 and TMPRSS2:ERG mRNA in EVs derived from readily obtained EPS urine constitute a viable clinical application for PCa screening [80].In order to evaluate available omics data regarding PCa and EVs we performed a detailed PubMed search with the terms displayed in Appendix A (performed in September, 2021). The number of retrieved studies is depicted in Figure 5. Our main interest was to scrutinize the 24 studies retrieved on the keywords proteomics, PCa and EVs excluding reviews (Figure 5A, depicted with a star). These 24 studies were manually curated for analysis and further discussion. Overall, our analysis suggests that proteomics and metabolomics studies in PCa are underrepresented compared to genomics and transcriptomics studies (Figure 5B).

The 24 retrieved publications on uEV proteome obtained from PCa patients are summarized in Appendix A. The manual curation of the 24 studies revealed that one study was wrongly annotated in the PubMed database as an experimental paper (Appendix A, red). Another study included all keywords in the title and abstract however was focused on bladder cancer instead (Appendix A, red). Of the remaining 22, eight publications described the clinical sample size, ranging from six to 107 subjects (Appendix A, depicted in green). Two of those were concerned with cataloging proteins, one aimed at identifying markers for resistance to docetaxel and five with identification of markers for csPCa. The majority of the studies were not statistically powered for biomarker discovery. Of the two studies with an acceptable clinical cohort size of approximately 100 subjects, one did not provide access to the mass spectrometry data [18] and the other used only mass spectrometry on PCa cell lines from which claudin 3 was identified as potential bio-marker [81]. Claudin 3 was then subsequently validated in a clinical cohort of 99 patients. From the eight clinical PCa EV studies only two provided public access to the mass spectrometry data. One provided mass spectrometry data from 12 patients [82] and the other with data obtained from cell lines [81]. In conclusion, there is only one study with publicly available MS data on uEVs from clinical samples with a modest cohort size of 12 patients [82]. It is our opinion that given the promising preliminary results on uEVs there is a need for a large scale PCa uEV proteomics profiling. Such a project will cast light on the diversity of uEV across patients, provide vital data for system biology analysis and provide more insight into the potential of uEV in diagnosis and prognosis of PCa. Furthermore, this can be facilitated by novel instrumentation with increased duty cycle enabling detailed proteome profiling based on short liquid chromatography gradients [83].

## 5. Proteomics in PCa

In the previous section, we discussed different biopsy sources and models for identification of potential protein EV biomarkers for diagnosis of PCa. Among these, more than half were based on PCa cell lines. Cell line based-biomarker discovery is a valuable approach; however, using large scale omics profiling several considerations should be in place. For example, one study has provided mass spectrometry (MS) data on EVs isolated directly from FBS and FBS depleted culture media [81]. We argue that this data set is an excellent resource to highlight potential false positive proteins in MS studies of EVs isolated from culture media. We consequently reanalyzed this specific subset of the data on EVs isolated from FBS media without human cells by searching against a human database including common MS contaminants such as keratin and bovine proteins. The search identified 954 human protein isoforms which corresponded to 463 unique protein coding genes as a result of homology between bovine and human proteins (Appendix A). Further, iBAQ values were estimated based on ion counts and log_2_ transformed to rank the proteins according to estimated iBAQ abundance. The top proteins included for example hemoglobin isoforms, tubulin, albumin, integrin beta, and Ras-related proteins. This protein list (Appendix A) will likely overestimate potential false positive protein hits since it is custom to deplete FBS, in for preparation of cell culture media in EV studies, for EVs isolation by ultracentrifugation. Nevertheless, we envisage that the abundance ranked list finds use in highlighting potential proposed markers that partly or fully originate from culture media proteins. As discussed in the previous section, EPS urine, seminal plasma and uEVs are based on noninvasive biopsies. From a clinical perspective EPS urine and uEVs are practical to collect on a large scale. In addition, paraffin embedded tissue samples are also typically readily available for protein biomarker studies.

### 5.1. Proteins Identified in PCa Proteomics Studies

State of art mass spectrometry facilitated the discovery of potential novel biomarkers for improved PCa diagnosis and prognostication. There is now extensive knowledge about tissue proteome, revealing proteomic features associated with malignant transformation as well as progression to metastasis.

Prostate is a gland producing a serous secretion which is rich in proteins. Prostate fluids that are collectable and clinically valuable are seminal plasma and expressed-prostatic secretion fluids. The seminal glands open into the prostatic urethra at its proximal side allowing collection of expressed-prostatic secretion fluids. Nevertheless, clinical collection in the voided urine following prostate massage is largely devoid of seminal vesicle derived proteins or sperm [77]. The prostatic fluid contains epithelial cells and secreted proteins. The secreted cells were recently analyzed on the molecular level by applying genomics and metabolomics analysis. In this setting, a detailed investigation of physiologic prostatic fluid showed differences between normal and pathologic protein prostate secretion [84]. MS analysis of prostatic secretion in urine identified 1022 proteins expressed after a prostate massage. Of these, 49 proteins were reported as enriched for prostate tissue. This list may serve to highlight biomarkers of prostate origin in urine-based studies [84] and represents a useful resource to match proteins identified in urine or uEVs as likely prostate origin (Appendix A). However, our intention is not to apply this list to restrict biomarker candidates since it is based on only 11 clinical samples which is unlikely sufficient to characterize the heterogeneity in PCa. Furthermore, the study was performed in 2012 and recent MS instrumentations are far more sensitive, questioning if a more complete protein list is obtainable with recent instrumentation.

Iglesias-Gato and co-workers performed a proteome profiling of 28 tumors and eight nonmalignant formalin-fixed paraffin-embedded (FFPE) radical prostatectomy specimens [85]. They applied super-SILAC by mixing FFPE protein extracts with isotopically labeled standard obtained from four prostate-derived cell lines. FFPE proteome profiling is less sensitive than fresh tissue-based proteomics, due to protein cross linking, but super-SILAC seems to partly mitigate this problem. Strong anion exchange chromatography was applied to simplify each sample into six fractions. The authors reported a total of 9000 protein identifications with a 1% FDR cut off. Whole-cell lysates are expected to result in considerably more protein identifications than FFPE samples. The authors report more than 5000 valid SILAC ratios for each sample. Proneuropeptide Y (Pro-NPY, C-terminal of NPY) expression, alone or in combination with the ERG status of the tumor, was associated with an increased risk of PCa specific mortality. In total, 649 differentially expressed proteins between malignant and nonmalignant were identified (Appendix A) [85]. Super-SILAC appears promising in terms of increasing the protein coverage and number of quantifiable proteins in clinical samples. A possibility is to explore super-SILAC for uEVs or EPS uEVs to increase coverage of such studies.

Seminal plasma is also considered for PCa protein biomarker studies and recently reviewed by Drabovich et al. [86]. However, seminal plasma proteins arise from secretions from the seminal vesicles which constitute about 65% of semen volume [86]. Moreover, semenogelins are highly abundant in seminal plasma which may compromise detection of low abundant proteins. Again, EVs are excellent for improving protein coverage by removing high abundant proteins in biofluids. For example, EVs in seminal plasma are mainly originating from prostate epithelial cells, and referred to as prostasomes [87]. By isolating seminal plasma EVs from vasectomized men, proteins were cataloged in two different EV size exclusion chromatography (SEC) fractions (thereby excluding contribution from testis and epididymis) [87]. Recently, a comprehensive multi omics approach targeting seminal plasma identified 76 candidate biomarkers and tested 19 proteins in seminal plasma of 67 negative biopsy and 152 PCa patients [88]. The Appendix A in the publication provided an extensive list of potential seminal plasma biomarkers and biomarker signatures for PCa obtained from different omics approaches and literature search [88]. The most significant regulated proteins obtained from clinical samples were extracted and added to Appendix A in this study. The authors specifically highlighted prostate-specific, secreted and androgen-regulated protein-glutamine gamma-glutamyltransferase 4 (TGM4) as a promising protein marker to include in biomarker panels [88]. Proteome profiling of seminal plasma EVs from 12 healthy donors resulted in a catalog of 1474 proteins [89]. Although seminal plasma appears as a promising liquid biopsy for PCa, it poses some clinical limitations in terms of sample collection on a large scale. In addition, to the analytical problems caused by abundant semenogelins.

PCa tissues representing the five grades, defined by the current Gleason Score classification system proposed by International Society of Urological Pathology, were quantified by label-free LC-MSMS [90]. This work provided tissue proteome characterization of five distinct PCa grades and benign prostate hyperplasia (BPH). The authors reported LMOD1, GYG1, IGKV3D-20, and RNASET2 as effective discriminators of low and high PCa grade group tissues. Furthermore, a panel of 11 prostate-derived proteins displayed the potential to stratify patients from low and high risk PCa (Appendix A). Tissue microarrays (TMAs) is an alternative method to direct LC-MS based studies. Tissue surface digestion and nano-LC-MS measurements can identify and quantify more than 500 proteins from a 1.5 mm diameter tissue section [91]. The authors highlighted 20 proteins which were the most significantly changed in expression between healthy and cancerous subjects based on 12 patients (Appendix A).

In in vitro models, Carvalho et al. [92] explored two public proteome data sets of the NCI-60 cancer cell lines panel, analyzing whole cell [93] and secreted EVs [93] proteomes. Proteins annotated to genes such as *SLC18A2*, *CPM*, *ATP6V1B1*, *HECTD1*, *GPN3*, *REG4*, *ZNF784*, *KIF4B*, *PKIA*, *RRAGB*, *PEMT*, *RNF115*, and *HAUS7* were uniquely identified in cellular proteomes of PCa compared to proteins identified in the other cell lines from the NCI 60 cancer cell line panel data set. In EVs, the proteins TRBC2, CAPN9, MSMP, UTS2B, and MYOZ2 were unique for PCa. These proteins differ from the other proteins listed in Appendix A by potentially differentiating PCa from other cancers rather than from healthy control samples. The authors concluded that cancer hallmarks proteins are contained in EVs in general and that protein content of EVs correlated with the cell of origin supporting the potential use of EVs as biomarkers (Appendix A).

Most of the Food and Drug Administration (FDA)-approved tumor markers are glycan antigens or glycoproteins [94,95,96]. The glycoproteome of PCa frozen tissues has also been targeted [42]. The glycopeptides were isolated by solid phase extraction and released by PNGase F followed by mass spectrometry analysis applying Sequential Window Acquisition of all Theoretical Mass Spectra (SWATH-MS). On average, 1430 N-glycosites were isolated from each sample and 220 glycoproteins displayed significant quantitative changes associated with PCa aggressiveness and metastasis. N-acylethanolamine acid amidase and protein tyrosine kinase 7 were suggested as potential biomarkers for aggressive PCa [42]. Scott and Munkley [97] provide an excellent review on the potential to exploit glycans as diagnostic and prognostic biomarkers for PCa. More specifically, glycoforms of PSA, sialylated glycans, O-GlcNAcylation and glycan branch structures are discussed as potential biomarkers for PCa [97]. Given the perceived relevance of glycans in cancer hallmarks, more studies on glycopeptides in clinical cohorts are expected as sample preparation methods and MS methodologies further develop.

Global protein profiles across large clinical cohorts are still limited in the literature and may provide valuable information regarding tumor heterogeneity. Furthermore, clinical studies targeting post translational modifications relevant in cancer on a large scale are limited. For example, methodologies for MS based profiling of acetylation [98], phosphorylation [99], ubiquitin [100], and ubiquitin like modifiers [101] start to reach a maturity that enables large scale profiling based on small sample amounts obtainable from liquid biopsies (e.g., phosphoproteomics by titanium oxide is possible based on even less than 100 micrograms of proteins as starting material).

In conclusion, the above mentioned proteomics studies provided novel biomarker candidates and improved our understanding of human physiology and molecular pathology of PCa by cataloging expressed proteins in tissue and subcellular compartments. In other words, these studies are justified by providing biomarker candidates and candidate therapeutic targets through system biology analysis of integrated analysis of data sets.

### 5.2. Comparing uEV Proteins from Different MS-Based Studies

This section focuses on uEV potential protein biomarkers obtained from MS-based studies. We refer to the recent review by Hatano et al. for discussion of potential biomarkers obtained from a broad range of technologies [102]. Previous studies have characterized EV proteins from PCa cells and identified annexin A2, calsyntenin 1, fatty acid synthetase, filamin C, folate hydrolase-1, and growth differentiation factor 15, which may be specific for PCa diagnosis [103]. Exportin-1 was also identified as a biomarker [74]. On the other hand, Notch3, milk fat globule epidermal growth factor 8, and inter-alpha-trypsin inhibitor heavy chain H4 were enriched in PCa EVs [104]. Khan et al. reported that exosomal survivin was a potential biomarker for early detection of PCa [105]. In addition, prostate cancer antigen 3 (PCA3), flotillin 2, Rab3B and late endosomal/lysosomal adaptor, MAPK and mTOR activator 1 (LAMTOR1) in EVs could be diagnostic markers for PCa [69,106].

Cancer initiation and progression are dependent on the ability of cells to communicate with their local and distant environment through secretory products such as EVs [107].

Analysis of uEVs enables the profiling of molecular changes that would be otherwise masked by the heterogeneity of whole urine. More than 50% of the proteins present in urine are also represented in vesicles as reported in studies comparing uEV proteome with whole urine [108,109]. In addition, analysis of the uEV proteome has provided information regarding the potential diverse functions of uEVs.

In this context, by profiling matched PCa tissue-derived extracellular vesicles and uEV, Dhondt et al. demonstrated that the uEV proteome is a reflection of the tissue of EV origin [82]. Their work presents a thoroughly substantiated case favoring the study of uEV as a source of unique biological and disease signatures not uncovered by the conventional analysis of crude urine samples. Overall, the authors have identified a total of 3686 proteins, which represents a more than twofold increase relative to previously published uEV proteomes. Among the identified proteins are PCa driver genes such as NKX3-1 and PTEN. Previously described PCa markers are FOLH1/PSMA, KLK3/PSA, androgen-regulated genes like FKBP5, and FAM129A [82]. Several proteins commonly overexpressed in PCa tissue were selectively enriched in uEV from PCa patients compared to post treatment. This was also the case when comparing PCa patients prior to treatment versus men with BPH [82]. The data provided by Dhondt et al. [82] is a thorough resource including uEV MS based proteomics data on prostate, bladder and renal cancer. However, a deep analysis of the global data was not provided in the original paper. We consequently reanalyzed the MS raw data files and performed multivariate analysis of all pairwise comparisons of the different provided sample types. The analysis was performed on iBAQ values [110] in which technical replicates were averaged before statistical analysis. The pairwise statistical analysis was calculated with the R package limma [111] and *p* values were adjusted by the method of Benjamini and Hochberg [112]. Potential biomarkers were extracted based on statistically significant regulated proteins from four pairwise comparisons between the samples pre-treatment, post-treatment, renal cancer, bladder cancer with uEVs from benign patients and compared with other proteins from PCa and PCa EV studies (Figure 6 and Appendix A). EV studies selected for inclusion in the comparison were based on the studies listed in Appendix A that included patient samples and aimed at PCa biomarker identifications [18,113,114]. Although, a list of proteins with suggested prostate specific proteins from Principe et al. were also included [84]. The PCa EV proteomics study by Fujita et al. [113], quantified 3528 proteins by deep proteomic analysis, which constitute one of the largest number of proteins ever reported in uEVs from patients with PCa, followed by validation of biomarker candidates by targeted-MS. The study aimed to discover biomarkers that might predict high-grade PCa and investigated the abundance of EV proteins isolated from DRE-urine in a cohort of six controls with negative-PCa biopsy, six low-grade PCa and six high-grade PCa patients using iTRAQ labeling LC-MS/MS.

Studies by Worst et al. [81] and Øverbye et al. [26] were not included in the comparison due to the limited list of proposed markers. However, both studies suggest Claudin-3 (CLDN3) and TMEM256 were additionally proposed by Øverbye et al. [26]. TMEM256 based classification resulted in an area under the receiver operating characteristic curve (AUC) of 0.87 [26]. The remaining studies included in the comparison were protein biomarker studies on various sources of biopsy samples aiming at identifying biomarkers for PCa [85,87,90,91]. It furthermore includes a list of proteins claimed by the authors to contain proteins enriched in prostate specific proteins [84].

Among the prognostically informative proteins, it is worth mentioning that CD63 is mainly associated with membranes of intracellular vesicles, although cell surface expression may also be induced. It has been shown that prostate basal epithelial cells do not express the characteristic CD antigens of secretory cells [115].

In conclusion, uEV are enriched in multiple PCa markers such as oncogenic drivers and androgen-regulated gene products. uEVs may provide additional insight into cancer-specific biological processes compared to the soluble urinary proteome. There is, in general, poor to no overlap between short listed validated protein markers for PCa across studies. However, comparing the full list of potential biomarkers from mass spectrometry across studies typically provides a significant overlap (Figure 6). Network analysis of PCa validated EV biomarkers are shared with potential biomarkers for bladder and renal cancer. Therefore, biomarkers for PCa must undergo scrutiny to assess specificity to PCa or urothelial cancers biomarkers.

In the paper by Dhondt et al. [82], results from functional analysis were presented without the consideration of the direction of regulation. Prostate, bladder and renal cancer patients present different responses to treatment. For example, immunotherapy is more efficient in bladder and renal and less effective in prostate cancer. Therefore, an overview of the regulated proteins function may provide novel therapeutic insight to functional differences between cancers. Therefore, we performed KEGG functional enrichment analysis of up regulated and down regulated proteins separately (Figure 7). The up regulated proteins in benign cases were enriched in proteasomes, and ribosomes compared to cancer cases (Figure 7a). Proteasomes and ribosomes are considered to co-purify in EV isolation and the data suggest that this co-purification is elevated for benign cases. On the other hand, proteins with functions like endocytosis, tight junction, and focal adhesion are elevated in the cancer uEVs (Figure 7b). We also observed general patterns of differences across cancer types where bladder and renal cancer stood out as more similar in terms of KEGG functional regulation compared to prostate cancer. The differences in KEGG functions need further research to elucidate if they contribute to difference in treatment response.

## 6. Therapeutic Significance of EV Proteins

EVs have biophysical properties, such as low toxicity and immunogenicity, stability, biocompatibility, and permeability, which are of vital importance to successful drug delivery systems. Furthermore, they have an enhanced circulation stability as well as bio-barrier permeation ability, therefore they can be used as effective chemotherapeutics carriers to improve the regulation of target tissues and organs [116].

In this context, EVs have the capacity to deliver different types of cargo and reach specific cells. EVs can be used as carriers to deliver therapeutic agents to tumor cells with the advantage of decreasing the side effects of drugs [56]. αvβ3 integrin holds promise as a non-invasive biomarker for PCa and its role in exosomes has a dominant effect on other pathways, as a potential therapeutic target [117,118].

EVs have also been shown, as mentioned above, to play a role in cell to cell communication by transferring biological material that may promote cancer progression and metastasis, which is another strong argument to further develop their potential as therapeutic targets in cancer [119,120,121].

Multi Drug Resistance (MDR) is one of the main limitations of cancer treatment. EVs potentially mediate multidrug resistance (MDR) through uptake of drugs in vesicles and thus limit the bioavailability of drugs to treat cancer cells. There is emerging evidence of the role EVs play in mediating drug resistance in advanced PCa. For other cancers, Sousa et al. [122] compared the RNA species present in drug-sensitive and MDR counterpart cells and in the EV’s released by those cells. They discovered two pseudogenes (a novel pseudogene and RNA 5.8S ribosomal pseudogene 2) as potential biomarkers for MDR.

Targeting EV proteins is of particular interest in cases of metastatic castration resistant prostate cancer (mCRPC), as patients frequently develop several metastatic sites that become resistant to treatment with anti-androgen deprivation therapy (ADT) [123]. Furthermore, Ishizuya et al. [123] identified novel therapeutic targets for CRPC by proteomic analysis of serum EVs. EVs were isolated, by ultracentrifugation, from sera from 36 men with metastatic PCa: untreated (*n* = 8), well-controlled with primary ADT (*n* = 8), and CRPC (*n* = 20). The authors have identified 823 proteins in the serum EVs. Six proteins were increased in CRPC patients compared with untreated patients. In contrast, only ACTN4 was increased in the CRPC patients compared to the ADT patients. ACTN4 was highly expressed in the PCa cell line DU145 and secreted EVs. RNA interference-mediated down regulation of ACTN4 significantly attenuated cell proliferation and invasion capacity of DU145 cells. Knockdown of ACTN4 effectively suppressed growth signaling pathways and invasive capacities of PCa cells, suggesting that blockade of ACTN4 may be a promising target therapy in CRPC patients [123].

## 7. Conclusions and Future Perspectives

In-depth research has been conducted towards the discovery of new biomarkers for diagnosis and prognosis of PCa due to the inability of current biomarkers to accurately predict disease aggressiveness.

Many studies confirm the potential of EVs as therapeutic vehicles for cancer treatment, according to its capacity to transfer cargos with both an immunoregulatory and genetic action. The lipid membrane of EVs makes them promising carriers of not only drugs but also other therapeutic molecules to target PCa. uEVs are a promising noninvasive and easily accessible source of biological material for investigation of biomarkers. MS-based proteomics enables large scale and deep profiling of uEV proteomes, which reflect the cellular processes associated with tissue-of-origin, creating new biological insights on PCa.

Future research is needed to unlock the potential of EVs in PCa diagnosis, prognosis, and therapy. We specifically suggest that uEVs from larger standardized clinical cohorts must undergo MS profiling to fully elucidate the heterogeneity in PCa. This will additionally enable systematic and meta-analysis of future proteomics studies. Furthermore, PCa studies must also evaluate samples from bladder and renal cancer to assess specificity for PCa. Finally, functional enrichment analysis suggested that uEV proteins from bladder and renal cancer are more similar on a functional level compared to prostate cancer.

## Figures and Tables

**Figure 1 ijms-22-13605-f001:**
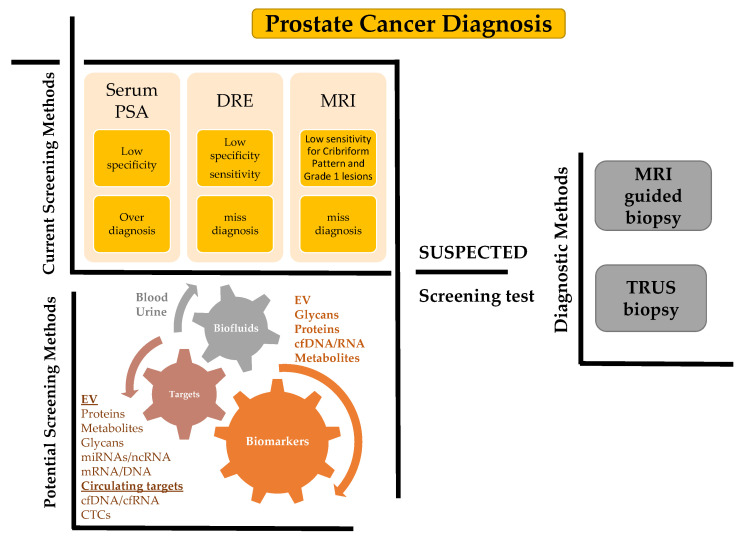
Systematic representation of the present and future possibilities for diagnosis of the PCa. The upper left panel highlights current limitations in screening methods. The lower left panel indicates future possibilities for EV based diagnostics and especially uEVs for PCa detection which will be further discussed in this study. The right panel indicates current diagnostics methods. CTCs: circulating tumor cells, cfDNA/cfRNA: cell free DNA/RNA, DRE: digital rectal examination, TRUS: transrectal ultrasound, and MRI: Magnetic Resonance Imaging.

**Figure 2 ijms-22-13605-f002:**
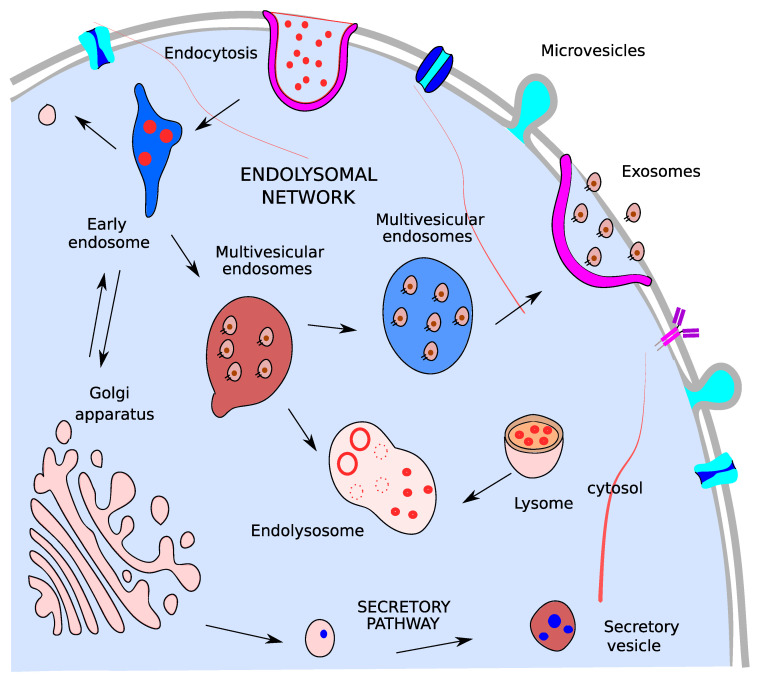
Exosomes biogenesis. Multivesicular endosomes in the cell which are produced by the invagination of endosomal limiting membrane and can secrete their content as exosomes or fuse with lysosomes to originating endolysomes.

**Figure 3 ijms-22-13605-f003:**
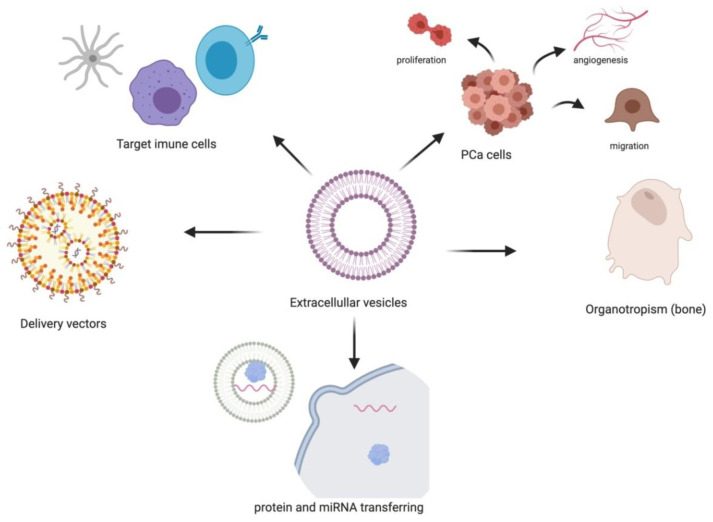
Functional effects described in the literature for PCa EVs.

**Figure 4 ijms-22-13605-f004:**
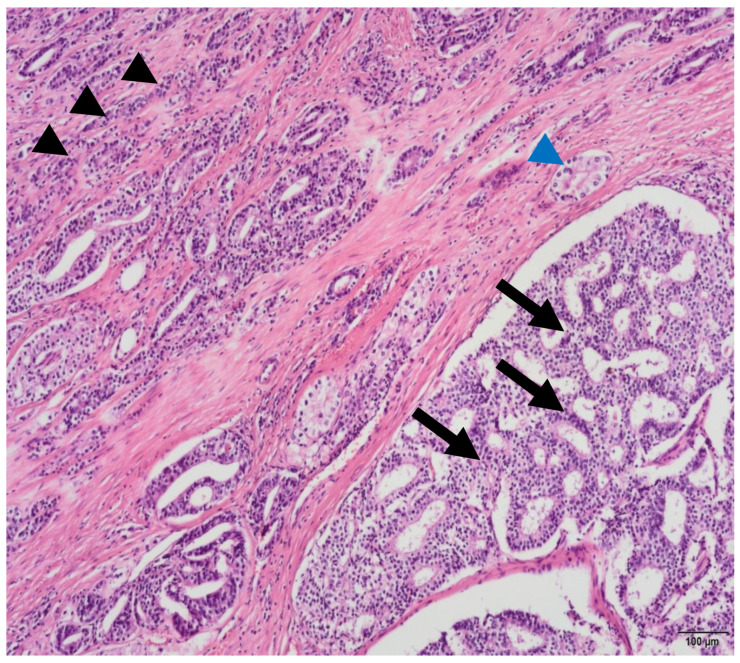
Photomicrograph of a prostatic acinar adenocarcinoma with multiple morphological patterns, such as some uniform medium sized glands (blue arrowhead), small poorly-formed and fused glands (black arrowhead) and both large and small round cribriform glands (arrows) with well-formed lumina (H&E, 10×). Image provided by João Pimentel MD, Pathology Department, Centro Hospitalar e Universitário Lisboa Central, October 2021.

**Figure 5 ijms-22-13605-f005:**
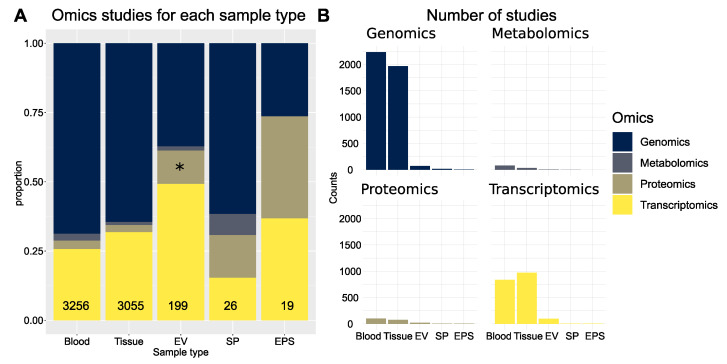
Overview of omics technologies applied in PCa using different biopsy samples. (**A**) The proportion of different omics technology based studies applied to each biopsy sample type are indicated on the y axis and the total numbers in the lower bars. (**B**) Number of retrieved PubMed studies in PCa for each omics technology and biopsy sample type. Abbreviations: expressed prostatic secretions (EPS), seminal plasma (SP) and extracellular vesicles (EV).

**Figure 6 ijms-22-13605-f006:**
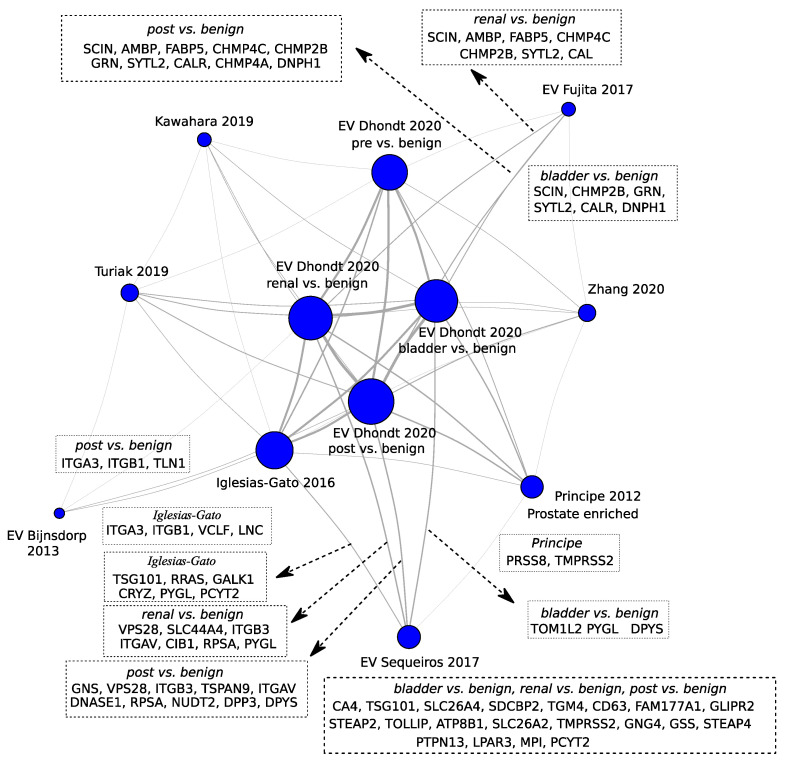
Network analysis depicting the proteins proposed as biomarkers by different studies in the intersections. The node sizes correlate with the number of potential biomarkers or prostate enriched proteins proposed in the studies: Kawahara 2019 [90], Dhondt 2020 [82], Fujita 2017 [113], Turiak 2019 [91], Zhang 2020 [87], Bijnsdorp 2013 [114], Iglesias-Gato 2016 [85], Principe 2012 [84], and Sequeiros 2017 [18]. The vertex thicknesses correlate with log_10_ of the number of proteins in the intersection. The actual proteins in the intersections were depicted for EV studies in which the proposed markers were validated. The term “pre” indicates samples from patients previous to treatment and “post” after treatment.

**Figure 7 ijms-22-13605-f007:**
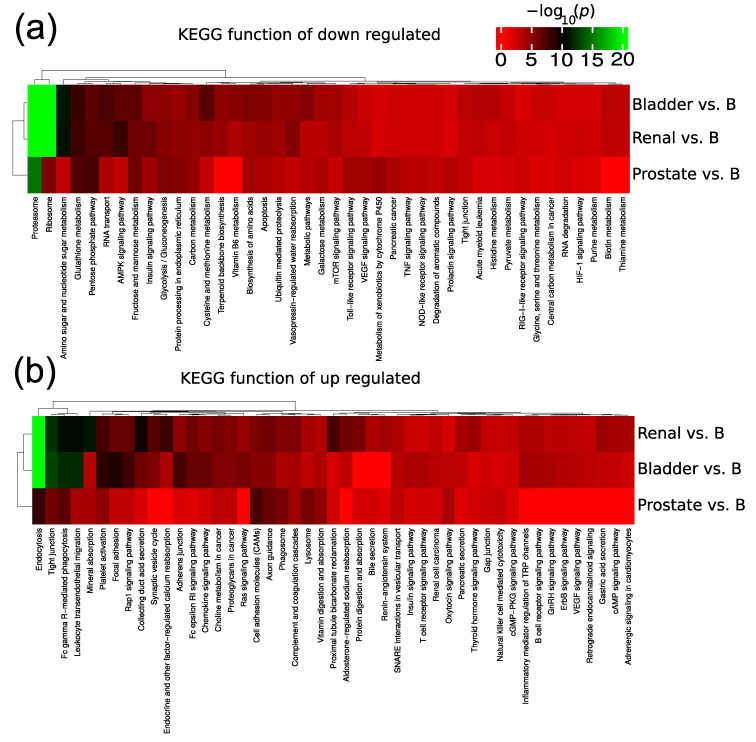
KEGG functional enrichment analysis of (**a**) significantly down- and (**b**) significantly up-regulated proteins in cancer compared to control cases. Significantly regulated proteins were defined as adjusted *p* value less than 0.05. B indicates benign.

**Table 1 ijms-22-13605-t001:** Reviews that provided a focused discussion on urine EVs’ application in PCa diagnosis.

Authors	Summary
Salciccia et al., 2021 [28]	biomarkers in PCa diagnosis including those derived from urine EVs compiled all clinical approved diagnostic kits for PCa
Drake and Kislinger, 2014 [29]	proteomics based PCa associated exosomes and their potential applications the EV proteome in PCa diagnosis
Nawaz et al., 2014 [30]	roles of extracellular vesicles in the most common urogenital cancers (prostate, kidney and bladder) EV isolation methods
Wang et al., 2020 [31]	exosome isolation techniques exosomal protein biomarker studies
Wu et al., 2019 [32]	EVs’ biophysical properties, roles and applications in the most common urologic neoplasms

## Data Availability

Not applicable.

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
