# Peer review of "Extracellular Vesicle Proteome in Prostate Cancer: A Comparative Analysis of Mass Spectrometry Studies"

_ijms, 2021, doi:10.3390/ijms222413605_

Round 1

Reviewer 1 Report

The authors aim to apply proteomics data of urine EVs and network analysis to identify potential biomarkers for PCa. They also propose that a larger standardized cohort potentially. However, the following problems need to be addressed.

  • Table 1 is incomplete, such as the author column. In addition, the authors should explain the article filter criteria because many related articles (urine EVs’ application in PCa diagnosis) can be found but not in the list. For example, “Extracellular vesicles in prostate cancer: a narrative review.”
  • Line 121~124, the authors mention that approved molecular diagnostic tests either measure mRNA expression or detection of gene fusion. miRNAs or methylation genes are also valuable biomarkers in urine EVs for the diagnosis of prostate cancer. The authors should describe comprehensively.
    • Extracellular vesicles in prostate cancer: a narrative review
    • Comparative Study of Extracellular Vesicles from the Urine of Healthy Individuals and Prostate Cancer Patients
  • In this review paper, if the authors could provide a complete experimentally verified protein list related to PCa and compare it with this research-identified biomarkers, they will help others cite this journal.
  • In “Comparing uEV proteins from different studies”, protein collection seems to be insufficient. From Table 1 of “Extracellular vesicles in prostate cancer: a narrative review”, a lot of proteins don’t discuss in this section.

Author Response

Response to reviewer 1

Reviewer 1:

The authors aim to apply proteomics data of urine EVs and network analysis to identify potential biomarkers for PCa. They also propose that a larger standardized cohort potentially. However, the following problems need to be addressed.

Authors: We thank the reviewer for the time invested in proving useful comments for improving the manuscript. We have below addressed each of the points put forward by the reviewer.

  • Table 1 is incomplete, such as the author column. In addition, the authors should explain the article filter criteria because many related articles (urine EVs’ application in PCa diagnosis) can be found but not in the list. For example, “Extracellular vesicles in prostate cancer: a narrative review.”

Authors: 

  • We noticed that our manuscript figures and tables were reformatted after we submitted the manuscript. Unfortunately, the tables lost some information after being reformatted. We contacted the editorial office to correct this issue, since this was not a problem in the submitted files. However, the manuscript was already submitted for review. We hope that this will not happen for our resubmitted manuscript.

2) The papers selected for in depth review were selected based on the search terms in supplemental table S1. In general, these search terms were aimed at targeting EV proteomics based mass spectrometry of prostate cancer samples. Specifically for table 1 as indicated in table caption, this list of reviews focus on urinary EVs application in PCa diagnosis. “Extracellular vesicles in prostate cancer: a narrative review.”, although excellent, discusses EVs from multiple liquid biopsies. We considered adding the suggested reference to table 1, but it would require a rephrasing of the table caption and open up for additional critics of left out studies.

Reviewer 1:

  • Line 121~124, the authors mention that approved molecular diagnostic tests either measure mRNA expression or detection of gene fusion. miRNAs or methylation genes are also valuable biomarkers in urine EVs for the diagnosis of prostate cancer. The authors should describe comprehensively.
    • Extracellular vesicles in prostate cancer: a narrative review
    • Comparative Study of Extracellular Vesicles from the Urine of Healthy Individuals and Prostate Cancer Patients

Authors:  Lines 121-124 are discussing approved diagnostic tests. The paper “Comparative Study of Extracellular Vesicles from the Urine of Healthy Individuals and Prostate Cancer Patients” is an experimental paper proposing potential biomarkers that has not been approved by health authorities.

Reviewer 1:

  • In this review paper, if the authors could provide a complete experimentally verified protein list related to PCa and compare it with this research-identified biomarkers, they will help others cite this journal.

Authors: Please see table S4 - we now include potential targets obtained by re-analysing data sets in Dhondt et al. by focusing on the prostate cancer. We also now include the functional analysis of the regulated proteins. To include experimental markers from other technologies than mass spectrometry would make the manuscript too broad, in our opinion.

  • In “Comparing uEV proteins from different studies”, protein collection seems to be insufficient. From Table 1 of “Extracellular vesicles in prostate cancer: a narrative review”, a lot of proteins don’t discuss in this section.

Authors:

The reviewer refers to section title “Comparing uEV proteins from different studies”. However, the title we have in the manuscript is “Comparing uEV proteins from different MS-based studies”. The review indicated by the reviewer also discusses proteins from ELISA and WB. We focus on experimentally verified proteins from mass spectrometry studies. Again, aiming for a complete coverage of proteins identified by all techniques will in our opinion make the review too broad and makes it almost impossible to fully cover in a single review text. We have nevertheless referenced the review indicated by the reviewer as a source for additional information on potential protein markers from a broad range of technologies. We have additionally modified the introduction of the section to clarify this issue.

Reviewer 2 Report

In this review the authors listed a number of manuscripts to compare the available mass spectrometry data regarding prostate cancer samples. The review is well-structured and has an informative comparison analysis of the available proteome content of PCa studies. The authors were keen in detailing the available studies and in criticizing some of the missing or non-public data of the mentioned studies. Nevertheless, some studies regarding protein content but not MS-based analysis might be left aside. In addition, some of the information present in this manuscript is not essential as it has already been extensively described, e.g the information about EV biogenesis, which makes this work very dense. The authors clearly describe the current methods for PCa detection and monitoring of the patients which highlights the necessity of finding novel biomarkers. Nevertheless, it is missing the clear comparison between analysing the whole body fluids vs EVs in the PCa context.

Reviewer suggestions:

  • It is not clear by the abstract if the authors are proposing a larger standardized cohort from urine EV samples or total body fluid analysis.

          It was also not clear what should give more specific/sensible results, if            the targeted plasma/serum EV studies or the urinary EV biomarkers.

  • In Figure 1, why is EVs underline in the bottom panel? It is strange as it suggests to be a title for what is described below. Nevertheless, CTC are not a subcategory of EVs.
  • There are no references for what the authors are describing between lines 93-100. Although what they are describing might be well established in the field, it would be appropriate to cite key references. Furthermore, a common and more recent subset of particles identified across EV preparation are exomeres, which were not mentioned (check for Zhang H, 2018, Nat Cell Bio). Are exomeres also present in urinary EVs?
  • In line 105, the word “pathologies” is not well applied. Pathology is the study of a certain disease. Not the disease itself.
  • In table 1, the column "authors" should be uniformed. E.g. in raw 1 there is the author name et al, year [reference]. In raw 2 there is only year [reference], in raw 4 there is only [reference], in the last raw there are no info on "authors" column. Furthermore, It is not clear the purpose of table 1.
  • “EE” abbreviation appears without description in line 149. Full name is missing at the first time in line 146. Same for ILV in line 150.
  • In line 101 the authors describe "Exosomes are 50-150 nm sized membrane vesicles". In line 153, the authors wrote "Exosomes range in size from 30–120 nm". There are several reports with differences in exosomes size. Nevertheless, in the same manuscript, the authors should be consistent.
  • In line 161, the authors describe the general content exosomes/EVs. Although, the authors should consider to include the presence of glycans in this description (please check manuscript Martins AM et al, Cells, 2021), especially because glycans as biomarkers are later explored and discussed by the authors.
  • Typo: Line 249 “… and relatively easy to be collected…”.
  • Please include in Figure 5 legend the abbreviation descriptions used in this figure.
  • Please include the reference of the “only study available” in line 319.
  • Typo: In line 409, “the” is repeated.
  • From lines 447-454 there are no references cited for what is been described.
  • It would be interesting to know from the studies mentioned in lines 473-477 if there was an advantage to look for uEV proteome in comparison with whole urine. In general, this would be an interesting novelty of this review.
  • t would help the readers if in Figure 6 the reference number of the cited studies would be provided rather than only the name of the first author, which is also not inclusive of all authors.
  • In line 557-558 where it is read "Diana et al (110) it should be Sousa et al (110)”.
  • In line 570 the authors should consider to correct "...PCa cell line DU1445 and in EVs from these cells.”.

Author Response

Response to reviewer 2

Reviewer 2:

In this review the authors listed a number of manuscripts to compare the available mass spectrometry data regarding prostate cancer samples. The review is well-structured and has an informative comparison analysis of the available proteome content of PCa studies. The authors were keen in detailing the available studies and in criticizing some of the missing or non-public data of the mentioned studies. Nevertheless, some studies regarding protein content but not MS-based analysis might be left aside. In addition, some of the information present in this manuscript is not essential as it has already been extensively described, e.g the information about EV biogenesis, which makes this work very dense. The authors clearly describe the current methods for PCa detection and monitoring of the patients which highlights the necessity of finding novel biomarkers. Nevertheless, it is missing the clear comparison between analysing the whole body fluids vs EVs in the PCa context.

Authors: We thank for the reviewer’s time and effort in providing constructive comments that helped us to improve the manuscript. We agree that EV biogenesis is already extensively described. We therefore aimed at keeping this section as short as possible. Consequently, we have now further shortened this section and referenced additional reviews on this topic. Furthermore, we now clearly argue why we prefer uEVs as a biomarker source.

Reviewer suggestions:

  • It is not clear by the abstract if the authors are proposing a larger standardized cohort from urine EV samples or total body fluid analysis.

Authors: This is a good point. We have now clarified these important statements in the updated abstract.

          It was also not clear what should give more specific/sensible results, if            the targeted plasma/serum EV studies or the urinary EV biomarkers.

Authors: Thanks - to make this clearer, we have updated the manuscript to clarify that our evaluation led to the conclusion that uEV gives the best proteome coverage and this together with release of biological material from prostate to urine makes urinary EVs the preferred biomarker source for proteomics. This is also reflected in the higher number of significant regulated proteins, even after correction of multiple testing, when comparing malignant versus benign.

  • In Figure 1, why is EVs underline in the bottom panel? It is strange as it suggests to be a title for what is described below. Nevertheless, CTC are not a subcategory of EVs.

Authors: We have corrected figure 1 accordingly.

  • There are no references for what the authors are describing between lines 93-100. Although what they are describing might be well established in the field, it would be appropriate to cite key references. Furthermore, a common and more recent subset of particles identified across EV preparation are exomeres, which were not mentioned (check for Zhang H, 2018, Nat Cell Bio). Are exomeres also present in urinary EVs?

Authors: To our knowledge exomers were first described by Zhang H, 2018 et al from cell line preparations and this has currently not been explored in biofluids. We have now added a discussion on this topic. We also added references to support the statements of this section.

  • In line 105, the word “pathologies” is not well applied. Pathology is the study of a certain disease. Not the disease itself.

Authors: Thanks corrected

  • In table 1, the column "authors" should be uniformed. E.g. in raw 1 there is the author name et al, year [reference]. In raw 2 there is only year [reference], in raw 4 there is only [reference], in the last raw there are no info on "authors" column. Furthermore, It is not clear the purpose of table 1.

Authors:  We noticed that our manuscript figures and tables were reformatted after we submitted the manuscript. Unfortunate, the tables lost some information after being reformatted. We contacted the editorial office to correct this issue, since this was not a problem in the submitted files. However, the manuscript was already submitted for review.  We hope that this will not happen for our resubmitted manuscript.

  • “EE” abbreviation appears without description in line 149. Full name is missing at the first time in line 146. Same for ILV in line 150.

Thanks - corrected

  • In line 101 the authors describe "Exosomes are 50-150 nm sized membrane vesicles". In line 153, the authors wrote "Exosomes range in size from 30–120 nm". There are several reports with differences in exosomes size. Nevertheless, in the same manuscript, the authors should be consistent.

Thanks - we have now removed the redundant section by shortening the biogenesis section

  • In line 161, the authors describe the general content exosomes/EVs. Although, the authors should consider to include the presence of glycans in this description (please check manuscript Martins AM et al, Cells, 2021), especially because glycans as biomarkers are later explored and discussed by the authors.

We fully agree that glycans and glycoproteins are relevant to highlight in this section, and we consequently modified the text according to the reviewer’s comment.

  • Typo: Line 249 “… and relatively easy to be collected…”.

Corrected

  • Please include in Figure 5 legend the abbreviation descriptions used in this figure.

Done

  • Please include the reference of the “only study available” in line 319.

We inserted the references and also some additional references in the sentence before to make the statements more transparent.

  • Typo: In line 409, “the” is repeated.

Corrected

  • From lines 447-454 there are no references cited for what is been described.

We inserted update references applying enrichment methods for the mentioned PTMs

  • It would be interesting to know from the studies mentioned in lines 473-477 if there was an advantage to look for uEV proteome in comparison with whole urine. In general, this would be an interesting novelty of this review.

We have compared the protein coverage obtained from void urine, urinary EVs, plasma and plasma EVs and discussed the results.

  • t would help the readers if in Figure 6 the reference number of the cited studies would be provided rather than only the name of the first author, which is also not inclusive of all authors.

We added the reference numbers to facilitate readability.

  • In line 557-558 where it is read "Diana et al (110) it should be Sousa et al (110)”.

Corrected

  • In line 570 the authors should consider to correct "...PCa cell line DU1445 and in EVs from these cells.”.

Thanks corrected

Round 2

Reviewer 1 Report

The title is “Extracellular vesicle proteome in prostate cancer: A comparative analysis”. It should be contained all of the techniques in molecular biologies, such as ELISA and WB, not just in MS-based studies. A lot of comment is about this. My final suggestion is to change the title or add content.

Author Response

We thank the reviewer for the comments and time invested. We agree with reviewer 1 that our title was not sufficiently precise in defining the scope of the manuscript. Consequently, we have changed the title according to the suggestion of the reviewer 1.

"Extracellular vesicle proteome in prostate cancer: A comparative analysis of mass spectrometry studies"